# Differential prevalence of pathobionts and host gene polymorphisms in chronic inflammatory intestinal diseases: Crohn's disease and intestinal tuberculosis

Imteyaz Ahmad Khan, Baibaswata Nayak, Manasvini Markandey, Aditya Bajaj, Mahak Verma, Sambudhha Kumar, Mukesh Kumar Singh, Saurabh Kedia, Vineet Ahuja⬚ *

Department of Gastroenterology and Human Nutrition Unit, All India Institute of Medical Sciences, New Delhi, India

* vineet.aiims@gmail.com

## Abstract

### Background and objectives

Crohn's disease (CD) and Intestinal tuberculosis (ITB) are chronic inflammatory ulcero-constrictive intestinal diseases with similar phenotype. Although both are disease models of chronic inflammation and their clinical presentations, imaging, histological and endoscopic findings are very similar, yet their etiologies are diverse. Hence, we aimed to look at differences in the prevalence of pathobionts like adherent-invasive *Escherichia coli (AIEC)*, *Listeria monocytogenes*, *Campylobacter jejuni* and *Yersinia enterocolitica* in CD and ITB as well as their associations with host-associated genetic polymorphisms in genes majorly involved in pathways of microbial handling and immune responses.

### Methods

The study cohort included 142 subjects (69 patients with CD, 32 with ITB and 41 controls). RT- PCR amplification was used to detect the presence of AIEC, *L. monocytogenes*, *C. jejuni*, and *Y. enterocolitica* DNA in colonic mucosal biopsies. Additionally, we tested three SNPs in IRGM (rs13361189, rs10065172, and rs4958847), one SNP in ATG16L1 (rs2241880) and one SNP in TNFRSF1A (rs4149570) by real-time PCR with SYBR green from peripheral blood samples in this cohort.

### Results

In patients with CD, AIEC was most frequently present (16/ 69, 23.19%) followed by *L. monocytogenes* (14/69, 20.29%), *C. jejuni* (9/69, 13.04%), and *Y. enterocolitica* (7/69, 10.14%). Among them, *L. monocytogenes* and *Y. enterocolitica* were significantly associated with CD (p = 0.02). In addition, we identified all the three SNPs in IRGM (rs13361189, rs10065172, and rs4958847), one SNP in ATG16L1 (rs2241880) and TNFRSF1A (rs4149570) with a significant difference in frequency in patients with CD compared with ITB and controls (p<0.05).

**Data Availability Statement:** All relevant data are within the manuscript and its Supporting information files.

**Funding:** Funding: The study was supported by a grant (30/3/2008/25-ECD-II) from the Indian Council of Medical Research (www.icmr.nic.in), New Delhi, India.

**Competing interests:** The authors have declared that no competing interests exist.

## Conclusion

Higher prevalence of host gene polymorphisms, as well as the presence of pathobionts, was seen in the colonic mucosa of patients with CD as compared to ITB, although both are disease models of chronic inflammation.

## Introduction

Crohn's Disease (CD) is a chronic, relapsing, transmural, inflammatory disorder of gastrointestinal tract, which results from the prolonged, uncontrolled immune response to pathogenic or commensal microflora, in genetically susceptible individuals. Though the etiology remains elusive, CD results from the complex interplay of host genetics and alterations in lifestyle or host environment [1]. These host-associated changes potentiate alterations in sensing and handling of gut commensals, which along with changes in structure and function of the gut microbial community, perpetuates the vicious cycle of gut dysbiosis and inflammation [2, 3].

Inflammatory bowel disease (IBD)-associated gut dysbiosis is characterized by a reduction in number and activity of protective gut commensals involved in the production of short-chain fatty acids, secondary bile acids and indole-based aryl-hydrocarbon receptor ligands [4–6]. The onset of toxic pro-inflammatory gut environment supports an overgrowth of entero-pathogens and functionally altered and potentially pathogenic commensal flora called the 'pathobionts'. Several of these pathogenic bacteria associated with CD, include adherent-invasive *Escherichia coli* (AIEC), *Listeria monocytogenes*, *Campylobacter jejuni*, *Mycobacterium avium* subspecies *paratuberculosis (MAP)* and *Yersinia enterocolitica* [7–11]. Even though an upsurge of numbers and pathogenic activities of these bacteria have been documented in CD, whether their expansion is one of the causes of the disease or is just an effect of changing gut environment remains undetermined.

Apart from the involvement of entero-pathogens and pathobionts, host genetics has also been highlighted to have a determinative impact in shaping disease etiology in CD. Genome-wide association studies (GWAS) have identified 240 genetic loci and single nucleotide gene polymorphisms (SNPs) which are associated with the risk of developing CD [12–14]. These studies have implicated various pathways involved in microbial handling and sensing and maintenance of epithelial barrier integrity, to be compromised in CD, further strengthening the role of microbial members as 'perpetuators' of gut inflammation. From the CD GWAS and meta-analyses, autophagy has been underlined as a key pathway implicated in CD etiology [15]. Autophagy is involved in antigen presentation, clearance of invading pathogens and secretion of antimicrobial peptides from the Paneth cells in the gut [16, 17]. SNPs in autophagy genes such as autophagy-related gene 16 like 1 (ATG16L1) and immunity-related GTPase family M (IRGM) have been associated with CD [15, 18–20]. Apart from the autophagy-mediated cellular innate immunity functions, GWAS studies have also highlighted other immune-associated functions accounting for the genetic susceptibility to CD. Tumour necrosis factor-alpha (TNFα), one of the proinflammatory cytokines, is established as a mediator of the inflammation in CD. TNFα receptor superfamily 1A (TNFRSF1A) encodes tumor necrosis factor receptor 1 (TNF-R1) and mutations in the gene can cause autoinflammatory disorders. Polymorphisms in TNFRSF1A have been studied for susceptibility, phenotypes and pharmacogenetics of CD. The TNFRSF1A- 609, G>T (rs4149570) has been shown to be associated with increased risk of CD [21–23].

The IBD burden is on the rise in tuberculosis endemic countries including India [1, 24]. Intestinal tuberculosis, a bacterial infection, presents itself as a chronic granulomatous disorder with phenotypic similarities with Crohn's Disease. This mimicry of clinical, radiological, endoscopic and histological manifestations, makes the differential disease diagnosis an enormous challenge [24]. Chronic inflammation is pathognomonic of both ITB and CD. Whether this chronic inflammation is a sole accountable factor for the ensued pathobiont bloom and gut dysbiosis or is attributable to disease-specific etiological events, is worth exploring.

Our recent study had shown significantly increased prevalence of MAP (23.2%, p = 0.03) in biopsy samples from patients with CD as compare to non-IBD controls [11]. The prevalence of key entero-pathogens and pathobionts namely adherent-invasive *Escherichia coli* (AIEC), *L. monocytogenes*, *C. jejuni*, and *Y. enterocolitica* in intestinal biopsy tissues of patients with CD, patients with ITB and non-IBD controls were not tested earlier. This study investigated the prevalence of these pathobionts and their association with single nucleotide genetic polymorphisms (SNPs) of IRGM, ATG16L1 and TNFRSF1A gene responsible for microbial sensing and handling in the CD and ITB patients and non-IBD control. The objective of this study is to compare the prevalence of bacterial entero-pathogens and SNPs in two diverse models of chronic immunoinflammatory granulomatous disease of the intestine: CD and ITB. The comparisons of these parameters to a bacterial infection of similar phenotypic presentation shall further consolidate the prevalence and potential role of these bacterial members and genetic polymorphisms in IBD.

## Materials and methods

### Study subjects

A total of 142 study subjects, including 101 patients with ulcero-constrictive disease of the ileo-colonic region and 41 patients with suspected haemorrhoidal bleed undergoing sigmoidoscopy served as controls, were recruited from the All India Institute of Medical Sciences, New Delhi, India. Among the 101 patients with ulcero-constrictive disease, 69 cases were diagnosed as CD (65.2%, Male), and 32 cases were diagnosed as ITB (62.5%, Male). Consecutive treatment naïve adult (age >18 yrs.) patients who have not received any immunomodulator or antitubercular therapy were included in this study. Patients previously treated with steroids; diagnosed with other autoimmune diseases, history of malignant tumour or complications were excluded. Diagnosis of CD and ITB were determined according to established guidelines based on standard clinical, radiological, endoscopic and histological criteria [24, 25]. Disease location and severity of CD were scored according to the Montreal classification [26]. The patients with ileocolonic transverse ulcers and/or strictures were diagnosed ITB with demonstration of caseating granulomas or acid fast bacilli or a positive culture on mucosal biopsies. The patients with presentation suggestive of ITB and concomitant active pulmonary tuberculosis were also included. The patients with diagnostic dilemma of ITB vs CD were given antitubercular therapy (ATT) trial for obtaining sustained response (clinical and mucosal healing). The patients achieved sustained clinical response at 6 months post-ATT were categorized as ITB and those do not respond to ATT but showed response to steroids or immunomodulators were categorized as CD. This study was approved by the institute ethics committee, All India Institute of Medical Sciences, New Delhi, India (Approval no. IEC/NP-165/2010). Written, informed consent was obtained from all patients and control subjects prior to study inclusion. All the samples were collected during the period 2011–2013. Study protocols were based on the ethical principles for medical research involving human subjects as per the Declaration of Helsinki.

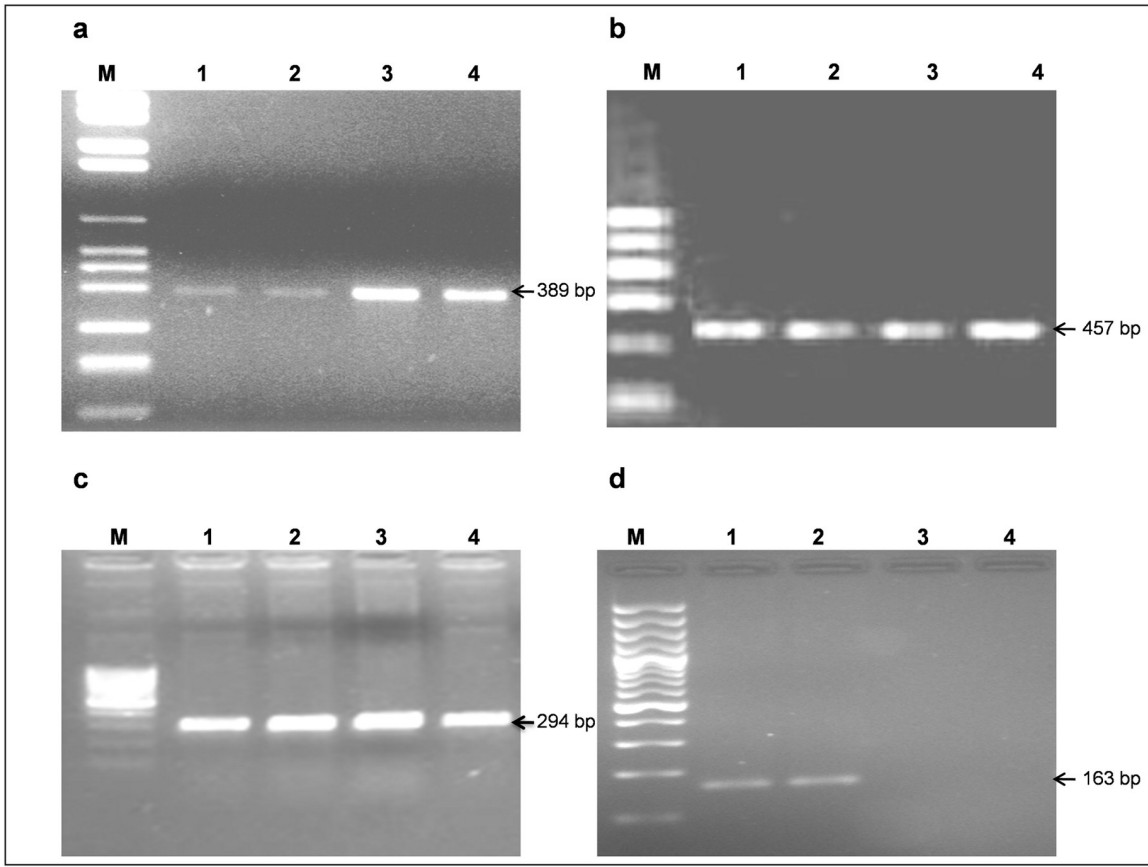

**Fig 1. Agarose gel electrophoresis of the RT-PCR products.** a) PCRs for AIEC. Lane M: Marker 100 bp; Lanes 1–4: AIEC positive samples; b) PCRs for *L. monocytogenes*. Lane M: Marker 200 bp; Lane 1: Positive control; Lanes 2–4: Positive samples; c) PCRs for *C. jejuni*. Lane M: Marker 100 bp; lane 1: Positive control; Lanes 2–4: Positive samples; d) PCRs for *Y. enterocolitica*. Lane M: Marker 100 bp; lane 1, 2: Positive samples.

## Genomic DNA isolation from intestinal biopsies for detection of pathogenic bacteria

Intestinal biopsies collected for detection of pathogenic bacteria were immediately snap frozen in liquid nitrogen and stored at—80˚C for isolation of genomic DNA later. Genomic DNA was isolated from intestinal biopsies (~15 mg) by commercial DNA extraction kit (DNeasy Blood & Tissue Kit, Qiagen, USA) using manufacturer's protocol. Detection of pathogenic bacteria (AIEC, *L. monocytogenes*, *C. jejuni* and *Y. enterocolitica*) were carried out by real-time PCR (Mx3005p, Agilent Technologies, USA) of genomic DNA isolated from intestinal biopsies using pathogen specific detection primer. These detection primers were designed for FimH, iap, 16S-23S ITS, and ail gene of bacteria (S1 Table). Briefly, real-time PCRs were carried out in 20-µl reaction volume containing 5 µl of DNA (~ 400 ng), 10 µl of Maxima SYBR Green qPCR Master Mix (Thermo Fisher Scientific), 1 µl of forward and 1 µl of reverse primers (20 pmol each), and 3 µL of nuclease-free water. Thermocycling conditions were initial denaturation at 95˚C for 5 minutes, followed by 35 cycles of denaturation (94˚C for 30 sec), annealing (mentioned temperature at S1 Table for 30 sec) and extension (72˚C for 30 s). Specific amplification was confirmed by plotting a dissociation curve through melting at the temperature range of 60˚C to 95˚C. The real-time PCR products for desired amplicon size were further confirmed by agarose gel (2%) electrophoresis (Fig 1).

## SNP genotyping of DNA isolated from peripheral blood

Genomic DNA was isolated from peripheral blood using the QIAmp DNA Blood kit (Qiagen, USA). The concentration and purity of DNA were determined by measuring absorbance at 260 nm and 280 nm. SNP genotyping was carried out for IRGM gene (rs13361189, rs10065172, and rs4958847), ATG16L1 gene (rs2241880), and TNFRSF1A gene (rs4149570) by real-time PCR using SYBR green-based chemistry. Allele-specific primers were designed and synthesized by introducing mismatches at the 3' terminal position of all forward primers (S2 Table). Briefly, SNP genotyping reactions were carried out in 20 μl volume containing 10 μl of Maxima SYBR Green qPCR Master Mix (Thermo Fisher Scientific), 1 μl (20 pmol) each forward and reverse primers and 3 μL of nuclease-free water. Allele-specific RT-PCR thermal cycling conditions were as follows: 95˚C for 5 min, followed by 40 cycles of denaturation (95˚C, 30 sec), annealing for 30 sec at respective annealing temperature for each primer set and extension (72˚C for 30 sec), followed by a melting curve analysis at 65˚C to 95˚C.

## Statistical analysis

Kruskal–Wallis and Mann-Whitney U tests were used to compare the prevalence of bacteria in 3 groups. Genotype and allele distributions among patients with CD and ITB versus healthy controls were compared using the $\chi2$ test or Fisher test as appropriate. Odds ratios (ORs) with a confidence interval (CI) of 95% were assessed to measure the strength of association. A chi-square test was used to analyze the deviation from Hardy-Weinberg equilibrium (HWE). Chi-square test *P*-value <0.05 was considered as an association. Association between the IRGM, ATG16L1, TNFRSF1A genotypes and bacterial positivity were assessed using unconditional logistic regression analysis. A *P*-value of less than 0.05 was considered statistically significant. Data were analyzed using STATA 14 software. Association of pathobiont prevalence and CD-associated SNP with the clinical variables were performed by Goodman and Kruskals (GK) Tau test using the GK tau data frame function of the Goodman Kruskal R package. This test determines the fraction of variability in one categorical variable that can be explained by the other categorical variable.

## Results

### Prevalence of *L. monocytogenes*, *Y. enterocolitica*, *C. jejuni* and AIEC in patients with CD, ITB and controls

Clinical characteristics of the study subjects are presented in Table 1. The presence of bacterial DNA in the mucosal biopsy samples of CD (n = 69), ITB (n = 32) and control (n = 41) subjects were detected by real-time PCR using pathogen-specific detection primers. Percentage of cases found positive for *L. monocytogenes* in CD, ITB and controls were 20.29% (14/69), 3.13% (1/32) and 7.32 (3/41) respectively. The prevalence of *L.monocytogenes* in patients with CD was statistically significant as compared to the ITB and control groups (p = 0.026) (Table 2). The prevalence of *Y. enterocolitica* in colonic biopsies of patients with CD was 10.14% (7/69), which was also significantly higher than ITB and controls (p = 0.02). We have also detected adherent invasive *Escherichia coli* (AIEC) and *C.jejuni* in CD, ITB and control subjects but the prevalence of these two pathogen were not found significant among groups (Table 2).

### Genotype and allele frequency distributions of SNPs in IRGM, ATG16L1 and TNFRSF1A gene among patients with CD and ITB

Genotyping by allele-specific real-time PCR for three SNPs in IRGM (rs13361189, rs10065172, and rs4958847), one SNP in ATG16L1 (rs2241880) and one SNP in TNFRSF1A (rs4149570)

**Table 1. Characteristics of the study population.**

| Variables | CD (n = 69) | ITB (n = 32) | Controls (n = 41) | P value |
|---|---|---|---|---|
| **Gender: (M/F)** | 45/24 | 20/12 | 29/12 | 0.7 |
| **Age at diagnosis** | 37.55 | 38.34 | 32.02 | 0.07 |
| **Mean duration of disease(months)** | 63.0 | 17.4 | | **0.001** |
| **Behaviour of disease (Montreal Classification)** | | | | |
| Non stricturing (B1) | 41 (59.4%) | 13 (40.6%) | | 0.1 |
| Stricturing (B2) | 27 (39.1%) | 19 (59.4%) | | |
| Penetrating (B3) | 1 (1.5%) | 0 | | |
| Perianal disease (P) | 0 | 0 | | |
| **Location of disease** | | | | |
| Ileal (L1) | 22 (31.9%) | 9 (28.1%) | | 0.9 |
| Colonic (L2) | 19 (27.5%) | 10 (31.3%) | | |
| Ileocolonic (L3) | 22 (31.9%) | 12 (37.5%) | | |
| Isolated upper digestive (L4) | 2 (2.9%) | 0 | | |
| L1+L4 | 1 (1.5%) | 1 (3.1%) | | |
| L2+L4 | 2 (2.9%) | 0 | | |
| L3+L4 | 1 (1.5%) | 0 | | |
| **Site of Biopsy** | | | | |
| Rectum | 1 (1.5%) | 0 | 2 (4.9%) | **0.001** |
| Rectosigmoid | 1 (1.5%) | 1 (3.1%) | 2 (4.9%) | |
| Sigmoid | 3 (4.4%) | 0 | 35 (85.4%) | |
| Descending Colon | 6 (8.7%) | 1 (3.1%) | 0 | |
| Transverse Colon | 6 (8.7%) | 3 (9.4%) | 0 | |
| Ascending Colon | 9 (13.0%) | 6 (18.8%) | 0 | |
| Caecum | 4 (5.8%) | 4 (12.5%) | 0 | |
| Ileocaecal | 13 (18.8%) | 10 (31.3%) | 0 | |
| Terminal Ileum | 26 (37.7%) | 7 (21.9%) | 2 (4.9%) | |

were carried out in total 142 subjects which included controls (n = 41) and patients as cases (CD, n = 69 and ITB, n = 32). Frequency of minor alleles and genotypes for each SNP in both cases (CD and ITB) and control populations are mentioned in Table 3. Expected genotypes were derived from observed genotypes frequency in case and controls by population genetic approaches. The test of deviations from Hardy-Weinberg equilibrium (HWE) in both case (CD, ITB) and control populations were evaluated by chi-square test using observed and expected genotype frequencies for each SNP. The test of deviation from HWE among control populations was not found significant (Table 3) which indicated that SNP loci were not influenced by evolutionary forces (mutation, genetic drift and migration) and suitable for genetic

**Table 2. Prevalence of bacteria isolated from biopsies of patients with CD, ITB and controls.**

| Name of the bacteria | CD (%) n = 69 | ITB (%) n = 32 | Controls (%) n = 41 | p-Value |
|---|---|---|---|---|
| *Adherent-invasive E. coli (AIEC)* | 16 (23.2) | 5 (15.6) | 9 (21.9) | 0.679 |
| *Listeria monocytogenes* | 14 (20.3) | 1 (3.1) | 3 (7.3) | **0.026** |
| *Campylobacter jejuni* | 9 (13.0) | 3 (9.4) | 2 (4.9) | 0.379 |
| *Yersinia enterocolitica* | 7 (10.1) | 0 | 0 | **0.02** |

**Table 3. Frequency and distribution of SNP allele and genotypes in cases (CD, ITB) and controls.**

| Gene/SNP | Genotype, Minor allele | CD (%) n = 69 | HWE P- value | ITB (%) n = 32 | HWE P-value | Controls (%) n = 41 | HWE P value |
|---|---|---|---|---|---|---|---|
| IRGM | CC | 10 (14.49) | **0.024** | 8 (25) | 0.307 | 17 (41.46) | 0.057 |
| rs13361189 | CT | 21 (30.43) | | 13 (40.6) | | 14 (34.15) | |
| (C/T) | TT | 38 (55.07) | | 11 (34.4) | | 10 (24.39) | |
| | C | 0.30 | | 0.45 | | 0.59 | |
| IRGM | CC | 14 (20.29) | **0.0002** | 12 (37.5) | 0.930 | 14 (34.15) | 0.594 |
| rs10065172 | CT | 17 (24.64) | | 15 (46.8) | | 19 (46.34) | |
| (C/T) | TT | 38 (55.07) | | 5 (15.6) | | 8 (19.51) | |
| | T | 0.67 | | 0.39 | | 0.44 | |
| IRGM | AA | 29 (42) | **0.001** | 8 (25) | 0.491 | 6 (14.63) | 0.677 |
| rs4958847 | AG | 21 (30.43) | | 14 (43.75) | | 21 (51.22) | |
| (A/G) | GG | 19 (27.54) | | 10 (31.25) | | 14 (34.15) | |
| | A | 0.57 | | 0.47 | | 0.40 | |
| ATG16L1 | CC | 34 (49.28) | **0.0002** | 5 (15.65) | 0.930 | 7 (17) | 0.815 |
| rs2241880 | CT | 18 (26) | | 15 (46.88) | | 19 (46.34) | |
| (C/T) | TT | 17 (24.64) | | 12(37.5) | | 15 (36.59) | |
| | C | 0.62 | | 0.39 | | 0.40 | |
| TNFRSF1A | GG | 20 (28.99) | **0.0001** | 14 (43.75) | 0.721 | 11 (26.83) | **0.049** |
| rs4149570 | GT | 17 (24.64) | | 15 (46.88) | | 26 (63.41) | |
| (G/T) | TT | 32 (46.38) | | 3 (9.38) | | 4 (9.76) | |
| | T | 0.59 | | 0.33 | | 0.41 | |

SNP = Single nucleotide polymorphism; CD = Crohn's disease; ITB = Intestinal tuberculosis; HWE = Hardy-Weinberg equilibrium

association studies. Among cases, HWE p-value was found significant for CD patients but not for ITB cases (Table 3).

## Genetic association of IRGM, ATG16L1 and TNFRSF1A gene SNPs with CD and ITB patients as compared to healthy controls

Genetic association of IRGM, ATG16L and TNFRSF1A gene SNPs were evaluated by estimating odds ratio (OR) of genotypes and alleles for case (CD and ITB) and healthy control populations. Dominant allele and genotypes for each SNP were assumed as reference or protective whereas, minor allele/genotypes were assumed as the risk for developing the disease as compared to healthy control. In our study population, the genetic association of above-mentioned gene SNPs was not observed for ITB cases as OR was not found significant (Table 4). The OR for risk allele/ genotypes was found significant (Table 4) for CD cases indicating the genetic association of IRGM, ATG16L and TNFRSF1A gene polymorphisms with the disease. The strength of association with significant OR >1 for each SNP indicated a risk for developing the disease, which was observed for the minor allele (T) and genotype (TT) of IRGM rs10065172; minor allele (A) and genotype (AA) of IRGM rs4958847 (A/G); minor allele (C) and genotype (CC) of ATG16L1 rs2241880 (C/T) and minor allele (T), genotype (TT and GT) of TNFRSF1A rs4149570 (G/T) (Table 4). The allele C was observed as the minor allele for IRGM SNP rs13361189 (C/T) in Indian CD patients. The OR <1 indicated protective for disease as shown for minor allele C indicating T as risk allele and TT as risk genotype (Table 4).

**Table 4. Association of risk allele and genotype of single nucleotide polymorphisms with CD and ITB cases as compared to healthy controls.**

| Gene/SNP | Genotype | CD OR (95% CI) | P-value HWE | ITB OR (95% CI) | P-value |
|---|---|---|---|---|---|
| IRGM rs13361189 (C/T) | TT | Ref | | Ref | |
| | CT | 0.39 [0.15–1.04] | 0.057 | 0.84 [0.27–2.64] | 0.771 |
| | CC | 0.15 [0.05–0.44] | **0.0002** | 0.42 [0.12–1.42] | 0.162 |
| | CC+CT | 0.26 [0.11–0.62] | **0.0017** | 0.61 [0.22–1.70] | 0.349 |
| | C | 0.29 [0.16–0.53] | **0.00003** | 0.58 [0.30–1.13] | 0.112 |
| IRGM rs10065172 (C/T) | CC | Ref | | Ref | |
| | CT | 0.89 [0.33–2.40] | 0.825 | 0.921 [0.33–2.57] | 0.875 |
| | TT | 4.75 [1.64–13.75] | **0.0029** | 0.72 [0.18–2.83] | 0.647 |
| | CT+TT | 2.03 [0.85–4.87] | 0.1067 | 0.86 [0.33–2.26] | 0.766 |
| | T | 2.77 [1.57–4.87] | **0.0003** | 0.86 [0.44–1.67] | 0.659 |
| IRGM rs4958847 (A/G) | GG | Ref | | Ref | |
| | AG | 0.73 [0.29–1.84] | 0.513 | 0.93 [0.32–2.68] | 0.898 |
| | AA | 3.56 [1.16–10.89] | **0.022** | 1.86 [0.49–7.08] | 0.356 |
| | AG+AA | 1.36 [0.59–3.14] | 0.464 | 1.14 [0.42–3.06] | 0.793 |
| | A | 1.98 [1.14–3.46] | **0.014** | 1.31 [0.67–2.53] | 0.422 |
| ATG16L1 rs2241880 (C/T) | TT | Ref | | Ref | |
| | CT | 0.83 [0.32–2.15] | 0.710 | 0.98 [0.35–2.72] | 0.930 |
| | CC | 4.28 [1.47–12.48] | **0.005** | 0.89 [0.22–3.53] | 0.871 |
| | CT+CC | 1.76 [0.76–4.08] | 0.182 | 0.96 [0.36–2.50] | 0.935 |
| | C | 2.45 [1.40–4.29] | **0.001** | 0.95 [0.48–1.85] | 0.884 |
| TNFRSF1A rs4149570 (G/T) | GG | Ref | | Ref | |
| | GT | 0.36 [0.13–0.93] | **0.033** | 0.45 [0.16–1.24] | 0.123 |
| | TT | 4.40 [1.23–15.72] | **0.017** | 0.58 [0.10–3.20] | 0.537 |
| | GT+TT | 0.89 [0.37–2.13] | 0.807 | 0.47 [0.17–1.25] | 0.130 |
| | T | 2.00 [1.15–3.49] | **0.013** | 0.68 [0.34–1.36] | 0.284 |

SNP = Single nucleotide polymorphism; CD = Crohn's diseae; ITB = Intestinal tuberculosis; HWE = Hardy-Weinberg equilibrium

## Association of IRGM, ATG16L1 and TNFRSF1A gene SNPs with bacterial infection susceptibility in CD patients

Increased prevalence of *L. monocytogenes* and *Y. enterocolitica* infection was observed in CD patients compared to ITB patients and control (Table 2). Association of IRGM, ATG16L1 and TNFRSF1A SNP with increased prevalence of these bacteria in CD patients was tested by chi-square test at 2 degrees of freedom for genotype and one degree of freedom for alleles. None of the SNPs in IRGM, ATG16L1 and TNFRSF1A gene was found to be significantly associated with susceptibility to bacterial infection (Table 5).

## Association of prevalent pathobionts with the clinical variables and group analysis

Goodman and Kruskal's Tau (GK τ) association measure between the pathobiont prevalence (noted as 'Positive' and 'Negative' for each pathobiont type) and the clinical parameters of patients with CD (age at onset of symptoms, age at which disease was diagnosed, location and behaviour of the disease) were plotted in Fig 2A. Association is indicated by the degenerate ellipse which is reflected as a straight line (GK τ = 1) for strong association and as full

**Table 5. Association of SNPs genotype with bacterial infection or persistence in CD patients.**

| | | AIEC | | L. monocytogenes | | C. Jejuni | | Y. enterocolitica | |
|---|---|---|---|---|---|---|---|---|---|
| | Genotype | Yes | No | Yes | No | Yes | No | Yes | No |
| IRGM rs13361189 (C/T) | CC | 1 | 9 | 1 | 9 | 0 | 10 | 0 | 10 |
| | CT | 7 | 14 | 5 | 16 | 2 | 19 | 2 | 19 |
| | TT | 8 | 30 | 2 | 36 | 7 | 31 | 5 | 33 |
| | χ2, df | 2.287, 2 | | 1.809, 2 | | 2.698, 2 | | 1.516, 2 | |
| | p value | 0.318 | | 0.404 | | 0.259 | | 0.468 | |
| IRGM rs10065172 (C/T) | CC | 3 | 11 | 4 | 10 | 2 | 12 | 1 | 13 |
| | CT | 3 | 14 | 2 | 15 | 2 | 15 | 3 | 14 |
| | TT | 10 | 28 | 8 | 30 | 5 | 33 | 3 | 35 |
| | χ2, df | 0.526, 2 | | 1.371, 2 | | 0.044, 2 | | 1.399, 2 | |
| | p value | 0.768 | | 0.503 | | 0.978 | | 0.496 | |
| IRGM rs4958847 (A/G) | AA | 8 | 21 | 6 | 23 | 4 | 25 | 3 | 26 |
| | AG | 4 | 17 | 4 | 17 | 1 | 20 | 1 | 20 |
| | GG | 4 | 15 | 4 | 15 | 4 | 15 | 3 | 16 |
| | χ2, df | 0.565, 2 | | 0.029, 2 | | 2.359, 2 | | 1.333, 2 | |
| | p value | 0.753 | | 0.985 | | 0.307 | | 0.513 | |
| ATG16L1 rs2241880 (C/T) | CC | 8 | 26 | 5 | 29 | 6 | 28 | 3 | 31 |
| | CT | 5 | 13 | 7 | 11 | 2 | 16 | 2 | 16 |
| | TT | 3 | 14 | 2 | 15 | 1 | 16 | 2 | 15 |
| | χ2, df | 0.508, 2, | | 5.269, 2 | | 1.463, 2 | | 0.132, 2 | |
| | p value | 0.775 | | 0.0717 | | 0.481 | | 0.935 | |
| TNFRSF1A rs4149570 (G/T) | CC | 4 | 16 | 5 | 15 | 2 | 18 | 1 | 19 |
| | CT | 4 | 13 | 4 | 13 | 3 | 14 | 2 | 15 |
| | TT | 8 | 24 | 5 | 27 | 4 | 28 | 4 | 28 |
| | χ2, df | 0.174, 2 | | 0.815, 2 | | 0.489, 2 | | 0.824, 2 | |
| | p value | 0.916 | | 0.665 | | 0.783 | | 0.662 | |

χ2, df = Chi-square, degrees of freedom

circle (GK τ = 0) for no association. The association measure between the pathobiont prevalence and clinical features failed to indicate any significant associations between the variables, except for a weak association between behaviour of disease (as per Montreal classification) and prevalence of *C.jejuni* (GK τ = 0.1) in patients with CD (Fig 2A). The GK τ association measure between the risk SNP genotype (noted as 'presence' and 'absence' for risk allele for each gene type) and the clinical features of the CD patients (age of disease onset, location and behaviour of the disease were plotted in Fig 2B. The location of disease showed weak association with the risk associated genotype in IRGM rs13361189 (GK τ = 0.17) and ATG16L1 rs2241880 (GK τ = 0.13), and a similar association was evident for the 'age at the onset of symptoms' with IRGM rs13361189 (GK τ = 0.11). No significant association was observed between risk loci and other clinical parameters. The pathobionts prevalence overlapping was observed by plotting the Venn diagram (Fig 2C) by using the 'venn' function of plots R package. Highest overlap was observed for the prevalence of AIEC and *L. monocytogenes* in patients with CD. This overlap of the two pathobionts was also evident from the Goodman and Kruskal's Tau value of 0.33, highlighting the co-occurrence of AIEC and *L.monocytogenes* in patients with CD (Fig 2A).

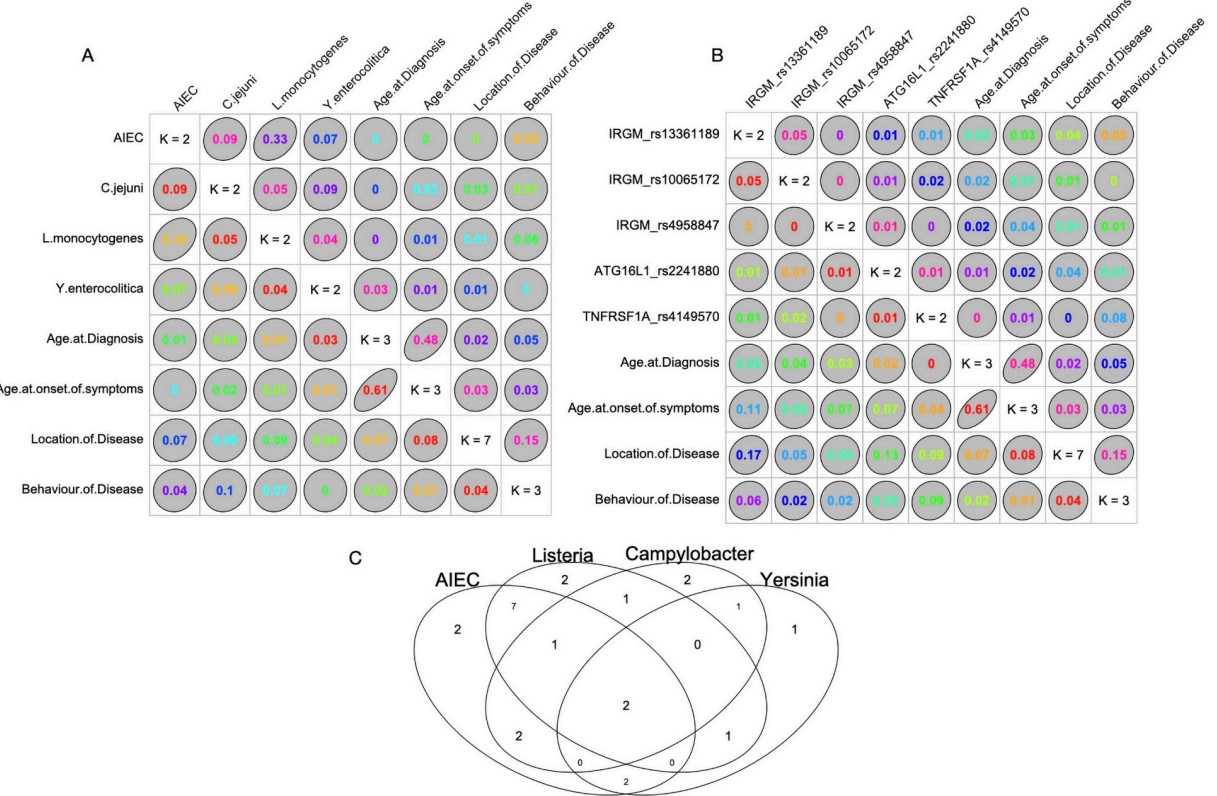

**Fig 2. Measurement of association and co-occurrence of the prevalence of pathobionts and risk alleles with the clinical features in patients with CD.** A) Association measurement between and amongst pathobiont prevalence (noted as 'Positive' and 'Negative' for each pathobiont type) and the clinical features of the patients with CD using Goodman and Kruskal's Tau association; B) Association measurement between the risk allele prevalence (noted as 'presence' and 'absence' for risk allele for each gene type) and amongst and the clinical features of the patients with CD using Goodman and Kruskal's Tau association; C) Venn Diagram depicting the co-occurrence of pathobionts in patients with CD.

## Discussion

Chronic gastrointestinal inflammation and disease-associated alterations in gut microenvironment act as stressors to drive disease-associated gut dysbiosis. Expansion of gut pathobionts and dwindling structure and function of beneficial members have been previously linked with CD [27, 28]. However, there is very less report about comparative assessment for the prevalence of various pathobionts in patients with CD and ITB. With an aim to explore the impact of chronic gastrointestinal inflammation in propelling a disease-associated bloom of major gut pathobionts, we have investigated the prevalence of *Yersinia enterocolitica*, *Listeria monocytogenes*, *Campylobacter jejuni*, and adherent-invasive *Escherichia coli* in two (CD and ITB) chronic intestinal inflammatory disease models of diagnostic dilemma. Simultaneously, the study also examined the prevalence of major risk SNP genotype/allele in the genes implicated for microbial sensing and handling such as IRGM (rs10065172, rs13361189 and rs4958847), ATG16L1 (rs2241880), and TNFRSF1A (rs4149570).

The prevalence of bacterial pathobiont was more in CD patients than ITB and control subjects. The control and ITB groups were found to have significant low infection incidences of *L. monocytogenes* and *Y. enterocolitica* as compared to CD patients whereas there was no significant difference in the incidences of the AIEC and *C. jejuni* among the groups. These observations are consistent with the study findings by Kang et al and Kallinowski et al respectively [4, 29]. Generally, the interconnection and exacerbations *between L. monocytogenes* and CD

endure to be an imperceptible dilemma in which more investigation is needed [9]. Although statistically insignificant, our results also reported enhanced prevalence of *C. jejuni* in patients with CD as compared to the ITB and control group. However, the prevalence of AIEC in patients with CD was similar to controls and, higher than ITB without any statistical insignificance. In a recent meta-analysis of 12 studies, the pooled prevalence of AIEC among patients with CD was 29% which is similar to our study, and in controls was 9%, which is much lower than our study [30]. This difference could be explained by the difference in patient population (only 1 study in this meta-analysis was from Asia (South Korea), where the prevalence of AIEC in controls was 22.2%, similar to our study), and analytical techniques, as in contrast to our study (which used PCR) these studies utilized adhesion and invasion assays. The two bacteria (*C.jejuni and AIEC*) have been reported to be prevalent in patients with CD and their pathogenic mechanisms have been implicated in the disease pathophysiology [10, 31]. Even though the sizes of the case and the control cohorts aid us to manifest the pathobiont expansion between the groups, a larger sample size would have helped to gain better statistical insights into our observations.

Despite the shared chronic inflammatory milieu in CD and ITB, the higher prevalence of these bacteria in CD stressed the explanatory power of CD-specific etiology in shaping the gut bacteria. CD is accompanied by an aberrant T-cell response, with an expansion of inflammatory Th1 and Th17 cell population and diminished gut T-regulatory cells, which in turn results in elevated oxygen radicals and nitric oxide. This remoulding of the gut environment might drive the expansion of inflammophilic gut pathobionts [32–35].

Human genetic association studies have enforced genetic contribution in the onset and progression of IBD. Genome-wide association studies (GWAS) have linked specific genetic polymorphisms with increased susceptibility to the development of IBD. The underlined genes and their products are important components of cellular pathways involved in, microbial sensing and handling, innate and adaptive immunity and maintenance of gut mucosal integrity.

Autophagy is involved in immune-specific functions including regulation of antigen presentation, secretion, inflammasome formation and mitophagy [36, 37]. Genetic polymorphisms in key autophagy genes like, ATG16L1 and IRGM have been linked to enhanced IBD susceptibility [15, 38]. The products of these genes actively regulate intracellular bacterial clearance in innate immune cells and may affect the structure of gut microbiome. IRGM rs13361189 minor allele carriers have been reported to have reduced expression of IRGM in whole blood and terminal ileum, along with altered expression of other genes associated with autophagy and inflammatory responses. Baskaran et al. have reported higher prevalence of risk SNPs genotype in the IBD patients of India. These SNPs also influence the disease pathophysiology by shaping the cellular microRNA milieu. IRGM rs10065172 has been reported to alter the binding site of miR-196 and downregulates the IRGM protective variant to dysregulate autophagy and intracellular bacterial handling process. Studies have shown that the CD-associated Thr300Ala mutation in ATG16L1 hampers the process of xenophagy, mediating an impaired efficiency of autophagy-mediated clearance of the intracellular enteric pathogen *Salmonella typhimurium*. Pierre et al. have shown the decreased efficiency of autophagy-mediated clearance of pathogenic adherent-invasive *Escherichia coli* (AIEC) in CD patients expressing the ATG16L1 variant [39]. In accordance with previous studies [15, 38, 40, 41], our results reveal a significantly enhanced prevalence of the risk alleles of IRGM (rs13361189, rs4958847 and rs10065172) and ATG16L1 (rs2241880) genes.

Tumor Necrosis Factor-alpha (TNF alpha) is an innate immune cytokine which mediates inflammation. TNF alpha acts as a ligand to the cell surface and membrane-bound receptors referred to as the TNF Receptor Superfamily 1A (TNFRSF1A) and plays a role in cell survival, apoptosis and inflammation. In our study, the TNFRSF1A polymorphism (rs4149570) was

found to be significantly associated with CD occurrence, thereby aligning with other studies, which highlight association of TNFRSF1A(rs4149570) genotype with increased risk of CD [23].

We could not demonstrate any association between the CD associated genetic polymorphisms and the prevalence of various pathobionts. Polymorphisms in NOD2 gene have been associated with AIEC colonization in patients with IBD [42, 43], however, we did not test for NOD2 in our cohort as previous studies have not found any association between NOD2 polymorphisms and Indian CD patients [44, 45].

The co-occurrence analysis of the pathobionts using the Goodman and Kruskal's Tau test, showed significant association between the prevalence of AIEC and *Listeria monocytogenes* in patient with CD. AIEC adhesion and invasion of intestinal epithelium in CD modulates the autophagy, which has been highlighted to enhance the survival of intracellular pathogens. One such mechanism by which AIEC dampens the autophagy process is through impairing SUMOylation process, which in turn controls intracellular survival of pathogens like *Listeria monocytogenes* in the gut epithelium [46]. Upon analysis of associations between the clinical parameters of patients with CD and prevalence of pathobionts and risk-associated genotypes, weak correlations were noted between the behaviour of disease (as per Montreal classification) and the prevalence of *C.jejuni*, and between the location of the disease and prevalence of risk variants of genes IRGM rs13361189 and ATG16L1 rs2241880. These associations are noteworthy and further investigations can yield interesting mechanistic insights.

Although these observations are encouraging, it is important to note the limitations. Firstly, the results must be validated with a larger cohort, keeping in mind that subjects represent wider ethnic and geographic backgrounds. Secondly, the genetic polymorphisms considered in this study are only restricted to genes which have already been implicated in CD. There might be other genes involved in microbial handling, which could promote the growth of pathobionts or decrease beneficial gut microbes when mutated. Therefore more such genetic loci must be taken into account.

## Conclusions

The present study highlights the differential prevalence of major gut pathobionts and genetic risk alleles in patients with CD and ITB. The study reveals that despite similar intestinal manifestations and chronic inflammation in CD and ITB, the complex disease-specific gut microenvironment is what determines the pathobiont inhabitation in the gut. The present study also discusses the prevalence of specific SNPs in genes implicated in IBD, in a North Indian cohort.

## Supporting information

**S1 Fig. The raw uncropped agarose gel, obtained post electrophoresis of the qPCR products.** (A) Amplicons corresponding to AIEC (Lanes 1–9). Lane M loaded with 100 bp DNA ladder. (B) Amplicons corresponding to *L.monocytogenes* (Lanes 2–6), Bands in lane 1 and lane M correspond to amplicon of positive bacterial culture and 200 bp ladder resp. (C) Amplicons corresponding to *C.jejuni* (Lanes 2–6), Bands in lane 1 and lane M correspond to amplicon of positive bacterial culture and 100 bp ladder resp. (D) Amplicons corresponding to *Y. enterocolitica* (Lanes 1 and 2), Bands in lane 5 and lane M correspond to amplicon of positive bacterial culture and 200 bp ladder resp.
(PDF)

**S1 Table. Primers used for detection of selected bacteria in the present study.** [a] F and R indicate forward and reverse primers, respectively.
(DOC)

**S2 Table. Allele-specific primers used for genotyping.** [a] F and R indicate forward and reverse primers respectively. [b] The polymorphic base is in bold; underlined letters indicate mismatched bases.
(DOCX)

## Author Contributions

**Conceptualization:** Imteyaz Ahmad Khan, Vineet Ahuja.

**Data curation:** Imteyaz Ahmad Khan, Baibaswata Nayak, Vineet Ahuja.

**Formal analysis:** Imteyaz Ahmad Khan, Baibaswata Nayak, Manasvini Markandey, Aditya Bajaj, Mahak Verma, Sambudhha Kumar, Mukesh Kumar Singh, Saurabh Kedia, Vineet Ahuja.

**Funding acquisition:** Vineet Ahuja.

**Investigation:** Imteyaz Ahmad Khan, Manasvini Markandey, Aditya Bajaj, Vineet Ahuja.

**Methodology:** Imteyaz Ahmad Khan, Vineet Ahuja.

**Project administration:** Vineet Ahuja.

**Supervision:** Vineet Ahuja.

**Validation:** Imteyaz Ahmad Khan, Vineet Ahuja.

**Writing – original draft:** Imteyaz Ahmad Khan, Vineet Ahuja.

**Writing – review & editing:** Baibaswata Nayak, Manasvini Markandey, Aditya Bajaj, Saurabh Kedia, Vineet Ahuja.

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
