## [Decision Letter · Decision Letter 0]

7 Jun 2021

PONE-D-21-12204

Differential prevalence of pathobionts and host gene polymorphisms in chronic inflammatory intestinal diseases: Crohn’s Disease and Intestinal Tuberculosis

PLOS ONE

Dear Dr. Ahuja,

Thank you for submitting your manuscript to PLOS ONE. After careful consideration, we feel that it has merit but does not fully meet PLOS ONE’s publication criteria as it currently stands. Therefore, we invite you to submit a revised version of the manuscript that addresses the points raised during the review process.

We look forward to receiving your revised manuscript.

Kind regards,

Santosh Chauhan, PhD

Academic Editor

PLOS ONE

Journal Requirements:

2)  In your Methods section, please provide additional information about the participant recruitment method and the demographic details of your participants. Please ensure you have provided sufficient details to replicate the analyses such as: a) the recruitment date range (month and year), b) a description of any inclusion/exclusion criteria that were applied to participant recruitment, c) a description of how participants were recruited.

3) Please ensure you have discussed the scientific rationale for the selection of the specific SNPs analysed in your genotyping analysis.

4) Please provide a sample size and power calculation in the Methods, or discuss the reasons for not performing one before study initiation.

5) PLOS ONE now requires that authors provide the original uncropped and unadjusted images underlying all blot or gel results reported in a submission’s figures or Supporting Information files. This policy and the journal’s other requirements for blot/gel reporting and figure preparation are described in detail at https://journals.plos.org/plosone/s/figures#loc-blot-and-gel-reporting-requirements and https://journals.plos.org/plosone/s/figures#loc-preparing-figures-from-image-files. When you submit your revised manuscript, please ensure that your figures adhere fully to these guidelines and provide the original underlying images for all blot or gel data reported in your submission. See the following link for instructions on providing the original image data: https://journals.plos.org/plosone/s/figures#loc-original-images-for-blots-and-gels.

6) Please include captions for your Supporting Information files at the end of your manuscript, and update any in-text citations to match accordingly. Please see our Supporting Information guidelines for more information: http://journals.plos.org/plosone/s/supporting-information.

Reviewers' comments:

Reviewer's Responses to Questions

**Comments to the Author**

1. Is the manuscript technically sound, and do the data support the conclusions?

Reviewer #1: Yes

Reviewer #2: Yes

2. Has the statistical analysis been performed appropriately and rigorously? 

Reviewer #1: Yes

Reviewer #2: Yes

3. Have the authors made all data underlying the findings in their manuscript fully available?

Reviewer #1: Yes

Reviewer #2: Yes

4. Is the manuscript presented in an intelligible fashion and written in standard English?

Reviewer #1: Yes

Reviewer #2: Yes

5. Review Comments to the Author

Reviewer #1: Overlapping clinical characteristics in Crohn’s Disease (CD) and intestinal tuberculosis (ITB) make the differential diagnosis a daunting task for clinicians, especially in case of patients from TB endemic countries like India. Differentiation between these two deadly diseases cannot be done with a standalone evaluation and requires a comprehensive approach which includes the sum of clinical, endoscopic, radiological, microbiological laboratory and culture studies for accurate diagnosis. Most of these approaches bear one or other limitations of low sensitivity and specificity. This dilemma results in misdiagnosis and delay in treatment contributing to increased morbidity and mortality. In this direction, study of both genetic and environmental risk factors may prove to be instrumental in deciphering newer biological markers for efficient diagnosis of CD and ITB. Dysbiosis in human gut microbiome and genetic susceptibility has been now associated with various diseases including CD. Thus, differential prevalence of pathobionts and host gene polymorphisms can serve as good biomarker to differentiate between CD and ITB.

The research article entitled “Differential prevalence of pathobionts and host gene polymorphisms in chronic inflammatory intestinal diseases: Crohn’s Disease and Intestinal Tuberculosis” (PONE-D-21-12204) by Khan et.al highlights the prevalence of pathobionts like adherent-invasive Escherichia coli (AIEC), Listeria monocytogenes, Campylobacter jejuni and Yersinia enterocolitica in Crohn’s Disease (CD) and Intestinal Tuberculosis (ITB). Consistent with previous studies, this study reveal a significantly enhanced prevalence of the risk alleles of IRGM (rs13361189, rs4958847 and rs10065172) and ATG16L1 (rs2241880) TNFRSF1A polymorphism (rs4149570) genes with CD.

The manuscript is well written and suitable statistical tools and analysis have been applied to determine the significance of the data. Although, the study is limited with lack of statistically significant data, possibly due to the small sample size as also admitted by authors in the manuscript, the findings from this study are very important especially the prevalence of specific SNPs in genes implicated in IBD, in a North Indian cohort. This study takes the field one step closer to a non-invasive and affordable diagnostic procedure for IBD and may prove to be useful for the follow up studies with a bigger sample size in Indian population. Such investigative studies should be pursued with further experiments involving the colonization of wild type, germ-free, and genetically modified mice with an individual bacterial species or with a combination of bacteria, in order to identify the exact causal bacterial strain or core microbiome and clarify the fate of the gut microbiota in IBD.

The manuscript may be considered for publication after addressing the following comments:

Comments:

1. In the introduction, authors mentioned Mycobacterium avium subspecies paratuberculosis (MAP) as one of the pathogen associated with CD. Also there is moderately high seroprevalence (23.4%) of MAP in human population of north India (SV Singh et.al., J Biol Sci 2014) so, adding MAP prevalence data in this study would have been useful and enhanced our understanding towards the association of MAP with CD in Indian population. Any specific reasons that authors didn’t included it in the study?

2. Refer lines 120-122 “Diagnosis of CD and ITB ………. based on standard clinical, radiological, endoscopic and histological criteria” Authors have provided data only from Behaviour of disease (Montreal Classification), location of disease and site of biopsy (refer Table 1). The other clinical, radiological histological, microbiological data is missing. Considering the overlapping clinical characteristics and knowing that the exclusive features between CD and ITB are caseation necrosis on biopsy, positive smear for acid-fast bacillus (AFB) and/or AFB culture, and necrotic lymph node on cross-sectional imaging in ITB (Kedia et.al.,2019), including supportive data from histology of biopsy samples (if possible) and clinical symptoms ( representative images of endoscopy of selected patients among all the groups) will be helpful in validation of the accurate diagnosis of the patients in this study. Alternatively, multivariable double logistic regression analysis of endoscopic and clinical features should have been performed. (Pls refer to research article by Li et.al., 2011, DOI 10.1007/s10620-010-1231-4).

3. Since sample storage conditions can affect the quantification of the target microbial markers especially fast growing E.coli, the storage conditions of the clinical samples (Biopsies) before the isolation of the nucleic acid should be described in the methods.

4. Refer lines 183-184 (C. jejuni prevalence) & 186-188 (AIEC prevalence). Since the Prevalence data for these two strains is not statistically significant, affirmative statements should be avoided. Sentences can be rewritten for better clarity and avoiding any misunderstanding.

5. Refer to lines 258-260, Pls provide suitable references of the previously published studies from the literature, if any, where the comparative assessment of the prevalence of various pathobionts, between CD and ITB was done. A comparison between their methods & findings with that in the present study will be useful.

6. Refer lines 269-270, The statement “The control and ITB groups were found to have low infection incidences” holds true only for L. monocytogenes and Y. enterocolitica as only these were significantly less prevalent in the control and ITB groups as compared to CD patients, whereas there was no significant difference in the incidence of the AIEC and C. jejuni between the groups. So sentence need to be modified accordingly.

7. Recent case study of an Asian female patient (Korean) with Crohn's Disease reported her case to be initially misdiagnosed as Intestinal Tuberculosis due to active pulmonary tuberculosis (Park et.al., Korean J Gastroenterol, 2021). Given that India is TB endemic country, it may be worthy to include the active pulmonary tuberculosis in the clinical history of the patient to avoid such misdiagnosis specifically in studies involving comparison of CD Vs. ITB. Also, since the baseline features of gut microbiota after or during anti-TB treatment among ITB patients may differ, it may be worth to mention the ATT treatment in recent past of subjects included in this study. The inclusion and exclusion criteria (especially history of tuberculosis, previous/existing anti-tubercular drug therapy at the time of specimen collection) for various groups (CD, ITB and Controls) under this study have not been defined and should be included in the manuscript.

8. As this study doesn’t demonstrate any association between the CD associated genetic polymorphisms and the prevalence of various pathobionts, its implications in pathogenesis of CD and ITB should have been discussed in correlation with clinical symptoms of patients under different groups in this study. This would have provided more insights to understand the role of these marker genes in etiology of CD and ITB.

Reviewer #2: The authors in this study aimed to look at differences in the prevalence of pathobionts like adherent-invasive Escherichia coli (AIEC), Listeria monocytogenes, Campylobacter jejuni and Yersinia enterocolitica in CD and ITB as well as their associations with host-associated genetic polymorphisms in genes majorly involved in pathways of microbial handling and immune responses. The study looks interesting; however, the authors should address the following concerns. The authors should also perform a general proof reading for typographical errors in the manuscript.

My comments are as below

1. The authors in this study aimed to look at the differences in the prevalence of various pathobionts as well as their associations with host-associated genetic polymorphisms in genes majorly involved in pathways of microbial handling and immune responses. Why were these 4 organisms chosen? It would have been better to perform a metagenome analysis to understand the profile of microorganisms in confirmed patients in a North Indian population.

2. Was the sample collection in the study prospective? It will be useful to include the details of the guidelines used for the classification of CD and ITB.

3. Genomic DNA was isolated from the intestinal biopsies. Kindly provide the details of the kit used. Why did the authors check the qPCR products on agarose gel electrophoresis? Usually a melt curve analysis is sufficient to prove specific amplification. What was the advantage of using qPCR? Was there any discrepancy between agarose gel profile and qPCR results?

Also, were the primers designed in the study or used from published literature? I am unable to see the point of final extension in real time PCR experiments, also the sizes of amplicons are too high for use in a qPCR assay. How did the authors confirm specificity of the assay? Were all samples performed with all assays simultaneously? What is the positive control?

4. Fig. 1-No amplicon sizes are marked here. This figure should not be a part of the main manuscript.

5. How did the allele specific PCR work in qPCR format? Was absence of amplification taken as presence of the SNP or was it done by melt curve analysis? Can the authors elaborate on the same?

6. The prevalence data for the organisms, are they overlapping? I think a grouped analysis should also be done.

7. The study reports interesting findings, but the authors should discuss about the link they expected to find between the presence of these specific pathobionts and genetic polymorphisms. The effect of these polymorphisms on the population should also be discussed. It will be interesting to understand the various polymorphisms associated with CD and ITB patients in this study. This will be evident in a grouped analysis i.e. in the 69 CD patients what all polymorphisms were significantly associated with CD, and not with ITB and controls.

The differential prevalence of major gut pathobionts and genetic risk alleles in patients with CD and ITB can be explored as a screening tool.

Minor

1. Line 82- remove ‘the’ before autophagy

2. TNF a is ‘TNFα’

3. Line 108, it is ITB not TB

4. Line 180 pathogen specific detection ‘primers’

5. Table 2 and 3, remove the term healthy control/HC. Replace with 'controls'

6. All tables, please provide the abbreviations in the footnote

7. Please provide reference for Montreal classification

8. Line 278, it should be statistical significance

9. All microorganisms’ names in italics in Fig. Legend

6. PLOS authors have the option to publish the peer review history of their article (what does this mean?). If published, this will include your full peer review and any attached files.

Reviewer #1: No

Reviewer #2: No

---

## [Author Response · Author response to Decision Letter 0]

22 Jul 2021

Ref: [PONE-D-21-12204] - [EMID: c79ffcd3e85c6421]

Response to the Editor’s comments

1) Please ensure that your manuscript meets PLOS ONE's style requirements, including those for file naming. The PLOS ONE style templates can be found at https://journals.plos.org/plosone/s/file?id=wjVg/PLOSOne_formatting_sample_main_body.pdf andhttps://journals.plos.org/plosone/s/file?id=ba62/PLOSOne_formatting_sample_title_authors_affiliations.pdf

2) In your Methods section, please provide additional information about the participant recruitment method and the demographic details of your participants. Please ensure you have provided sufficient details to replicate the analyses such as: a) the recruitment date range (month and year), b) a description of any inclusion/exclusion criteria that were applied to participant recruitment, c) a description of how participants were recruited.

Ans: Inclusion and exclusion criteria, methodology, demographic details have been mentioned in the text (Ref line-128-133, 136-143).

Among the 101 patients with ulcero-constrictive disease, 69 cases were diagnosed as CD (65.2%, Male), and 32 cases were diagnosed as ITB (62.5%, Male). Consecutive treatment naïve adult (age >18 yrs) patients who have not received any immunomodulator or antitubercular therapy were included in this study. Patients previously treated with steroids; diagnosed with other autoimmune diseases, history of malignant tumor or complications were excluded.

3) Please ensure you have discussed the scientific rationale for the selection of the specific SNPs analysed in your genotyping analysis.

Ans: The rationale behind selection of the specific SNPs has been expanded in the “Discussion” section of the manuscript (Line No 301-310). 

“However, there is very less report about comparative assessment for the prevalence of various pathobionts in patients with CD and ITB. With an aim to explore the impact of chronic gastrointestinal inflammation in propelling a disease-associated bloom of major gut pathobionts, we have investigated the prevalence of Yersinia enterocolitica, Listeria monocytogenes, Campylobacter jejuni, and adherent-invasive Escherichia coli in two (CD and ITB) chronic intestinal inflammatory disease models of diagnostic dilemma. Simultaneously, the study also examined the prevalence of major risk SNP genotype/allele in the genes implicated for microbial sensing and handling such as IRGM (rs10065172, rs13361189 and rs4958847), ATG16L1 (rs2241880), and TNFRSF1A (rs4149570)”. 

4) Please provide a sample size and power calculation in the Methods, or discuss the reasons for not performing one before study initiation.

Ans: The present study is an extension to our earlier paper published by our group in 2016 (Khan IA, Pilli S, A S, Rampal R, Chauhan SK, Tiwari V, Mouli VP, Kedia S, Nayak B, Das P, Makharia GK, Ahuja V. Prevalence and Association of Mycobacterium avium subspecies paratuberculosis with Disease Course in Patients with Ulcero-Constrictive Ileocolonic Disease. PLoS One. 2016 Mar 28; 11(3):e0152063. doi: 10.1371/journal.pone.0152063. PMID: 27019109; PMCID: PMC4809507). Our earlier study was analysed for the prevalence of Mycobacterium avian subspecies paratuberculosis (MAP) in patients with Crohn’s disease and intestinal tuberculosis. In this present study, we have utilizedsame samples for the prevalence of other four pathobionts and genetic association with risk SNP genotypes/ alleles of the genes implicated for microbial sensing and handling. Constraints for sample size and power calculation occur due to the exploratory nature of this study to carry out with the available samples. 

5) PLOS ONE now requires that authors provide the original uncropped and unadjusted images underlying all blot or gel results reported in a submission’s figures or Supporting Information files. This policy and the journal’s other requirements for blot/gel reporting and figure preparation are described in detail at https://journals.plos.org/plosone/s/figures#loc-blot-and-gel-reporting-requirements and https://journals.plos.org/plosone/s/figures#loc-preparing-figures-from-image-files. When you submit your revised manuscript, please ensure that your figures adhere fully to these guidelines and provide the original underlying images for all blot or gel data reported in your submission. See the following link for instructions on providing the original image data: https://journals.plos.org/plosone/s/figures#loc-original-images-for-blots-and-gels.

Ans: Provided

6) Please include captions for your Supporting Information files at the end of your manuscript, and update any in-text citations to match accordingly. Please see our Supporting Information guidelines for more information: http://journals.plos.org/plosone/s/supporting-information.

Ans: Captions for Supporting Information is included at the end of the manuscript. 

Response to Reviewers’ Comments:

Reviewer #1: Overlapping clinical characteristics in Crohn’s Disease (CD) and intestinal tuberculosis (ITB) make the differential diagnosis a daunting task for clinicians, especially in case of patients from TB endemic countries like India. Differentiation between these two deadly diseases cannot be done with a standalone evaluation and requires a comprehensive approach which includes the sum of clinical, endoscopic, radiological, microbiological laboratory and culture studies for accurate diagnosis. Most of these approaches bear one or other limitations of low sensitivity and specificity. This dilemma results in misdiagnosis and delay in treatment contributing to increased morbidity and mortality. In this direction, study of both genetic and environmental risk factors may prove to be instrumental in deciphering newer biological markers for efficient diagnosis of CD and ITB. Dysbiosis in human gut microbiome and genetic susceptibility has been now associated with various diseases including CD. Thus, differential prevalence of pathobionts and host gene polymorphisms can serve as good biomarker to differentiate between CD and ITB.

The research article entitled “Differential prevalence of pathobionts and host gene polymorphisms in chronic inflammatory intestinal diseases: Crohn’s Disease and Intestinal Tuberculosis” (PONE-D-21-12204) by Khan et.al highlights the prevalence of pathobionts like adherent-invasive Escherichia coli (AIEC), Listeria monocytogenes, Campylobacter jejuni and Yersinia enterocolitica in Crohn’s Disease (CD) and Intestinal Tuberculosis (ITB). Consistent with previous studies, this study reveal a significantly enhanced prevalence of the risk alleles of IRGM (rs13361189, rs4958847 and rs10065172) and ATG16L1 (rs2241880) TNFRSF1A polymorphism (rs4149570) genes with CD.

The manuscript is well written and suitable statistical tools and analysis have been applied to determine the significance of the data. Although, the study is limited with lack of statistically significant data, possibly due to the small sample size as also admitted by authors in the manuscript, the findings from this study are very important especially the prevalence of specific SNPs in genes implicated in IBD, in a North Indian cohort. This study takes the field one step closer to a non-invasive and affordable diagnostic procedure for IBD and may prove to be useful for the follow up studies with a bigger sample size in Indian population. Such investigative studies should be pursued with further experiments involving the colonization of wild type, germ-free, and genetically modified mice with an individual bacterial species or with a combination of bacteria, in order to identify the exact causal bacterial strain or core microbiome and clarify the fate of the gut microbiota in IBD.

The manuscript may be considered for publication after addressing the following comments:

Reviewer 1

Comment1: In the introduction, authors mentioned Mycobacterium avium subspecies paratuberculosis (MAP) as one of the pathogen associated with CD. Also there is moderately high seroprevalence (23.4%) of MAP in human population of north India (SV Singh et.al., J Biol Sci 2014) so, adding MAP prevalence data in this study would have been useful and enhanced our understanding towards the association of MAP with CD in Indian population. Any specific reasons that authors didn’t included it in the study? 

Response 1: We have included prevalence of MAP as reported by us in the introduction section. The section was modified as follows (Line 108-115)

Our recent study had shown significantly increased prevalence of MAP (23.2%, p=0.03) in biopsy samples from patients with CD as compare to non-IBD controls (Khan PLoS One. 2016). The prevalence of key enteropathogens and pathobionts namely adherent-invasive Escherichia coli (AIEC), L. monocytogenes, C. jejuni, and Y. enterocolitica in intestinal biopsy tissues of patients with CD, non-IBD controls and patients with ITB were not tested earlier. This study investigated the prevalence of these pathobionts and their association with single nucleotide genetic polymorphisms (SNPs) of IRGM, ATG16L1 and TNFRSF1A gene in CD and ITB patients as compare to non-IBD control.

Comment 2: Refer lines 120-122 “Diagnosis of CD and ITB ………. based on standard clinical, radiological, endoscopic and histological criteria” Authors have provided data only from Behaviour of disease (Montreal Classification), location of disease and site of biopsy (refer Table 1). The other clinical, radiological histological, microbiological data is missing. Considering the overlapping clinical characteristics and knowing that the exclusive features between CD and ITB are caseation necrosis on biopsy, positive smear for acid-fast bacillus (AFB) and/or AFB culture, and necrotic lymph node on cross-sectional imaging in ITB (Kedia et.al.,2019), including supportive data from histology of biopsy samples (if possible) and clinical symptoms ( representative images of endoscopy of selected patients among all the groups) will be helpful in validation of the accurate diagnosis of the patients in this study. Alternatively, multivariable double logistic regression analysis of endoscopic and clinical features should have been performed. (Pls refer to research article by Li et.al., 2011, DOI 10.1007/s10620-010-1231-4). 

Response 2: We have considered other clinical, radiological histological, microbiological parameters for diagnosis of ITB and CD which are mentioned in the methodology section. However, intestinal biopsies were collected at baseline for diagnosis of CD/ITB from naïve cases who had not received any immunomodulator therapy or antitubercular therapy. Briefly, diagnostic criteria is mentioned in the methodology section as follows and the same has been mentioned in the text as follows (line-136- 143)

“The patients with ileocolonic transverse ulcers and/or strictures were diagnosed ITB with demonstration of caseating granulomas or acid fast bacilli or a positive culture on mucosal biopsies.The patients with presentation suggestive of ITB and concomitant active pulmonary tuberculosis was also included. The patients with diagnostic dilemma of ITB vs CD were given antitubercular therapy (ATT) trial for obtaining sustained response (clinical and mucosal healing). The patients achieved sustained clinical response at 6 months post-ATT were categorized as ITB and those do not respond to ATT but showed response to steroids or immunomodulators were categorized as CD.”. 

Comment 3: Since sample storage conditions can affect the quantification of the target microbial markers especially fast growing E.coli, the storage conditions of the clinical samples (Biopsies) before the isolation of the nucleic acid should be described in the methods. 

Response 3: We have included in the text Line 148-153. The text reads as follows:

Intestinal biopsies collected for detection of pathogenic bacteria were immediately snap frozen in liquid nitrogen and stored at - 80°C for isolation of genomic DNA later. Genomic DNA was isolated from intestinal biopsies (~15 mg) by commercial DNA extraction kit (DNeasy Blood & Tissue Kit, Qiagen, USA) using manufacturer’s protocol.

Comment 4: Referlines 183-184 (C. jejuni prevalence) & 186-188 (AIEC prevalence). Since the Prevalence data for these two strains is not statistically significant, affirmative statements should be avoided. Sentences can be rewritten for better clarity and avoiding any misunderstanding. 

Response 4: As suggested, we have modified the sentence as follows (Line 207-209)

We have also detected adherent invasive Escherichia coli (AIEC) and C.jejuni in CD, ITB and control subjects but the prevalence of these two pathogen were not found significant among groups (Table 2).

Comment 5: Refer to lines 258-260, Pls provide suitable references of the previously published studies from the literature, if any, where the comparative assessment of the prevalence of various pathobionts, between CD and ITB was done. A comparison between their methods & findings with that in the present study will be useful. 

Response 5: Corrections have been made in the “Discussions” section of the manuscript.

 Line-301-302

Comment 6: Refer lines 269-270, The statement “The control and ITB groups were found to have low infection incidences” holds true only for L. monocytogenes and Y. enterocolitica as only these were significantly less prevalent in the control and ITB groups as compared to CD patients, whereas there was no significant difference in the incidence of the AIEC and C. jejuni between the groups. So sentence need to be modified accordingly. 

Response 6: As suggested we have changed the sentence in the Discussion section. Line-311-314.

The prevalence of bacterial pathobiont was more in CD patients than ITB and control subjects. The control and ITB groups were found to have significant low infection incidences of L. monocytogenes and Y. enterocolitica as compared to CD patients whereas there was no significant difference in the incidences of the AIEC and C. jejuni among the groups.

Comment 7: Recent case study of an Asian female patient (Korean) with Crohn's Disease reported her case to be initially misdiagnosed as Intestinal Tuberculosis due to active pulmonary tuberculosis (Park et.al., Korean J Gastroenterol, 2021). Given that India is TB endemic country, it may be worthy to include the active pulmonary tuberculosis in the clinical history of the patient to avoid such misdiagnosis specifically in studies involving comparison of CD Vs. ITB. Also, since the baseline features of gut microbiota after or during anti-TB treatment among ITB patients may differ, it may be worth to mention the ATT treatment in recent past of subjects included in this study. The inclusion and exclusion criteria (especially history of tuberculosis, previous/existing anti-tubercular drug therapy at the time of specimen collection) for various groups (CD, ITB and Controls) under this study have not been defined and should be included in the manuscript. 

Response 7: Details have been added to the relevant paragraph in the Methods section.

Ref line-136-143

The patients with ileocolonic transverse ulcers and/or strictures were diagnosed ITB with demonstration of caseating granulomas or acid fast bacilli or a positive culture on mucosal biopsies.The patients with presentation suggestive of intestinal TB and concomitant active pulmonary tuberculosis were also included. The patients with diagnostic dilemma of ITB vs CD were given antitubercular therapy (ATT) trial for sustained response (clinical and mucosal healing) to ATT. The patients with sustained clinical response at 6 months post-ATT were categorized as intestinal TB and those do not respond to ATT but shows response to steroids or immunomodulators were categorized as Crohn’s disease patients.

Comment 8: As this study doesn’t demonstrate any association between the CD associated genetic polymorphisms and the prevalence of various pathobionts, its implications in pathogenesis of CD and ITB should have been discussed in correlation with clinical symptoms of patients under different groups in this study. This would have provided more insights to understand the role of these marker genes in etiology of CD and ITB. 

Response 8: As suggested we reanalyze the data for association of CD-associated genetic polymorphism and the pathobiont prevalence with the clinical symptoms of patients using Goodman and Kruskals Tau test. The methodology is added in the line 190-193.The association data is mentioned in the result section along with Figure 2 a, b.

Association of pathobiont prevalence and CD-associated SNP with the clinical variables were performed by Goodman and Kruskals Tau test using the GKtaudataframe function of the GoodmanKruskal R package. This test determines the fraction of variability in one categorical variable that can be explained by the other categorical variable.

Figure 2a shows the results of the Goodman and Kruskal’s Tau association measure between the pathobiont prevalence (noted as ‘Positive’ and ‘Negative’ for each pathobiont type) and the clinical features of the patients with CD (age of disease onset, location and behaviour of the disease). The association measure between the pathobiont prevalence and clinical features indicate that location of disease explains variability in L.monocytogenes and AIEC prevalence in patients with CD (GK τ = 0.33). 

Figure 2b shows the results of the Goodman and Kruskal’s Tau association measure between the risk allele prevalence (noted as ‘presence’ and ‘absence’ for risk allele for each gene type) and the clinical features of the patients with CD (age at disease diagnosis, age at onset of symptoms, location and behaviour of the disease). The location of disease showed weak association with the risk associated genotype in IRGM rs13361189 (GK �=0.17) and ATG16L1 rs2241880 (GK �=0.13), and a similar association was evident for the ‘age at the onset of symptoms’ with IRGM rs13361189 (GK �=0.11). No significant association was observed between risk loci and other clinical parameters.

Reviewer 2

 Reviewer #2: The authors in this study aimed to look at differences in the prevalence of pathobionts like adherent-invasive Escherichia coli (AIEC), Listeria monocytogenes, Campylobacter jejuni and Yersinia enterocolitica in CD and ITB as well as their associations with host-associated genetic polymorphisms in genes majorly involved in pathways of microbial handling and immune responses. The study looks interesting; however, the authors should address the following concerns. The authors should also perform a general proof reading for typographical errors in the manuscript.

Comment1: The authors in this study aimed to look at the differences in the prevalence of various pathobionts as well as their associations with host-associated genetic polymorphisms in genes majorly involved in pathways of microbial handling and immune responses. Why were these 4 organisms chosen? It would have been better to perform a metagenome analysis to understand the profile of microorganisms in confirmed patients in a North Indian population.

Ans 1: The pathobionts analyzed in the study were selected on the basis of their previously reported positive association with the occurrence of Crohn’s Disease. The vicious cycle of gut pathobiont bloom and persistent inflammation has been highlighted to be an important aspect of inflammatory bowel disease. The metagenome analysis for understanding the community-level bacterial composition will be an important addition to the study and shall be attempted in the next phase. However, with the present aim to investigate the role of complex disease-specific gut micro-environment in supporting the pathobiont inhabitation in the gut, 16S-based, marker gene analysis shall undermine the prevalence of the pathobionts of interest, and would have added to the costs of sequencing and analysis. 

Comment 2: Was the sample collection in the study prospective? It will be useful to include the details of the guidelines used for the classification of CD and ITB.

Ans 2: The samples analyzed in the present study were collected prospectively and the detail guidelines were included in the methodology section.

Comment 3: Genomic DNA was isolated from the intestinal biopsies. Kindly provide the details of the kit used. 

Ans 3: Kit detail has been added. Line- 151-152. (DNeasy Blood & Tissue Kit, Qiagen, USA).

Comment 4: Why did the authors check the qPCR products on agarose gel electrophoresis? Usually a melt curve analysis is sufficient to prove specific amplification.

Ans 4: During real time PCR, melt curve analysis step was included to visualize nonspecific amplification. At end reaction, we have rechecked RT-PCR product for amplicon of desired size and agarose gel photographs were taken(Fig.1). 

Comment 5: What was the advantage of using qPCR? 

Ans 5: We have done qualitative real time PCR using SYBR green chemistry. We have chosen this method over conventional PCR for detection low copy number of target DNA. Allele Specific Real time PCR assay is well established rapid protocol for SNP genotyping of large number of samples at same time. 

Comment 6: Was there any discrepancy between agarose gel profile and qPCR results?

Ans 6: The amplified RT-PCR product corresponded with the predicted amplicon size. However concentration of amplicon or band visualization in the agarose gel was inversely proportional to the Ct value of the real time PCR. 

Comment 7: Also, were the primers designed in the study or used from published literature?

Ans 7: The allele-specific primers used in this study were designed using the Primer 3 software and bacterial primers used from published literature (reference added in the Supplementary Table 1).

Comment 8: I am unable to see the point of final extension in real time PCR experiments; also the sizes of amplicons are too high for use in a qPCR assay. 

Ans 8: There was no final extension step in the real time PCR. So corrections have been made in the text (methodology section). In Taqman based qPCR assay, amplicon size is usually 100bp or small amplicon size. In our study, we have used SYBR Green chemistry based qualitative real time PCR where amplicon sizes varied from 110-457 bp. This size of amplicon can be amplified in RT-PCR without affecting Ct value. 

Comment 9: How did the authors confirm specificity of the assay? 

Ans 9: The pathogen specific primer specificity was initially checked by conventional PCR using bacterial culture. In real time PCR, melt curve analysis was done and further reaction end products were checked in agarose gel for desired amplicon size. 

Comment 10: Were all samples performed with all assays simultaneously? 

Ans 10: All genomic DNA isolated from intestinal biopsies was tested simultaneously using pathogen specific detection primer set. Allele specific SNP genotyping PCR were performed in two separate tube (biallelic) using template genomic DNA isolated from peripheral blood. 

Comment 11: What is the positive control?

Ans 11: Positive controls for bacteria obtained from lab culture identified previously.

Comment 12: Fig. 1-No amplicon sizes are marked here. This figure should not be a part of the main manuscript.

Ans 12: Now amplicon sizes are marked in gel picture. We want to keep this figure in the manuscript because it represents pathogen detection and the amplicon of desired sizes when amplified by pathogen specific detection primer set. 

Comment 13: How did the allele specific PCR work in qPCR format? Was absence of amplification taken as presence of the SNP or was it done by melt curve analysis? Can the authors elaborate on the same?

Ans 13: For allele specific PCR, we have designed one common reverse primer and two (wild/mutant type) forward primers with one nucleotide mismatch at the end following polymorphic site. Two tubes PCR method using wild or mutant type primer set were used for genotyping. There will be no amplification when there is mismatch allele. The polymorphisms were confirmed either as the presence or absence of amplification with or without Ct value.

Comment 14: The prevalence data for the organisms are they overlapping? I think a grouped analysis should also be done.

Ans 14: The grouped analysis was performed for the prevalent pathobionts. 

Figure 2c shows the Venn diagram describing the overlap in the prevalence of the pathobionts. Highest overlap can be observed in the prevalence of AIEC and L.monocytogenes in patients with CD. Figure 2a Goodman and Kruskal’s Tau association measure across the prevalence of various pathobionts indicate the similar co-occurence relationship between the AIEC and L.monocytogenes prevalence.

Comments 15: The study reports interesting findings, but the authors should discuss about the link they expected to find between the presence of these specific pathobionts and genetic polymorphisms. The effect of these polymorphisms on the population should also be discussed. It will be interesting to understand the various polymorphisms associated with CD and ITB patients in this study. This will be evident in a grouped analysis i.e. in the 69 CD patients what all polymorphisms were significantly associated with CD, and not with ITB and controls.

Response 15: First of all, we studied association of genetic polymorphism with CD or ITB. We did not find any association of these polymorphisms with ITB but association was observed with CD. Then we have studied whether these polymorphisms impaired clearance or favor growth of pathobionts in the risk allele/genotype. Association risk genotypes with bacterial persistence in CD patients were studied. Due to small sample size we did not find any significant association. However, association of clinical variable with CD-associated genetic polymorphism and prevalent pathobiont were observed in Goodman and Kruskals Tau test and grouped analysis. Large cohort study will give deep insight to it. Small number size is the limitation of our study.

Comment 16: The differential prevalence of major gut pathobionts and genetic risk alleles in patients with CD and ITB can be explored as a screening tool.

Ans: Yes, we will plan to explore it as screening tool. Gut microbiome study is ongoing in our laboratory and broader knowledge will help us to design screening tool. There is genetic predisposition for CD and these SNPs can be explored as screening tool for risk assessment. 

The following Corrections have been made in the revised version of the manuscript.

Minor

1. Line 82- remove ‘the’ before autophagy

Ans: Corrected

2. TNF a is ‘TNFα’

Ans: Corrected

3. Line 108, it is ITB not TB

Ans: Corrected

4. Line 180 pathogen specific detection ‘primers’

Ans: Corrected

5. Table 2 and 3, remove the term healthy control/HC. Replace with 'controls'

Ans: Corrected

6. All tables, please provide the abbreviations in the footnote

Ans: Corrected

7. Please provide reference for Montreal classification

Ans: Corrected

8. Line 278, it should be statistical significance

Ans: Corrected

9. All microorganisms’ names in italics in Fig. Legend

Ans: Corrected

---

## [Decision Letter · Decision Letter 1]

2 Aug 2021

Differential prevalence of pathobionts and host gene polymorphisms in chronic inflammatory intestinal diseases: Crohn’s Disease and Intestinal Tuberculosis

PONE-D-21-12204R1

Dear Dr. Ahuja,

We’re pleased to inform you that your manuscript has been judged scientifically suitable for publication and will be formally accepted for publication once it meets all outstanding technical requirements.

Kind regards,

Santosh Chauhan, PhD

Academic Editor

PLOS ONE

Additional Editor Comments (optional):

Reviewers' comments:

Reviewer's Responses to Questions

**Comments to the Author**

1. If the authors have adequately addressed your comments raised in a previous round of review and you feel that this manuscript is now acceptable for publication, you may indicate that here to bypass the “Comments to the Author” section, enter your conflict of interest statement in the “Confidential to Editor” section, and submit your "Accept" recommendation.

Reviewer #1: All comments have been addressed

Reviewer #2: All comments have been addressed

2. Is the manuscript technically sound, and do the data support the conclusions?

Reviewer #1: Yes

Reviewer #2: (No Response)

3. Has the statistical analysis been performed appropriately and rigorously? 

Reviewer #1: Yes

Reviewer #2: (No Response)

4. Have the authors made all data underlying the findings in their manuscript fully available?

Reviewer #1: Yes

Reviewer #2: (No Response)

5. Is the manuscript presented in an intelligible fashion and written in standard English?

Reviewer #1: Yes

Reviewer #2: (No Response)

6. Review Comments to the Author

Reviewer #1: The resolution of Fig 2a looks poor and text in the image looks pixelated, a good resolution image should be provided.

Reviewer #2: (No Response)

7. PLOS authors have the option to publish the peer review history of their article (what does this mean?). If published, this will include your full peer review and any attached files.

Reviewer #1: **Yes: **Atul Vashist

Reviewer #2: No

---

## [Editor Report · Acceptance letter]

9 Aug 2021

PONE-D-21-12204R1 

Differential prevalence of pathobionts and host gene polymorphisms in chronic inflammatory intestinal diseases: Crohn’s Disease and Intestinal Tuberculosis 

Dear Dr. Ahuja:

I'm pleased to inform you that your manuscript has been deemed suitable for publication in PLOS ONE. Congratulations! Your manuscript is now with our production department. 

Kind regards, 

on behalf of

Dr. Santosh Chauhan 

Academic Editor

PLOS ONE